# PAC-learning in the presence of evasion adversaries

**Daniel Cullina**
Princeton University
dcullina@princeton.edu

**Arjun Nitin Bhagoji**
Princeton University
abhagoji@princeton.edu

**Prateek Mittal**
Princeton University
pmittal@princeton.edu

## Abstract

The existence of evasion attacks during the test phase of machine learning algorithms represents a significant challenge to both their deployment and understanding. These attacks can be carried out by adding imperceptible perturbations to inputs to generate *adversarial examples* and finding effective defenses and detectors has proven to be difficult. In this paper, we step away from the attack-defense arms race and seek to understand the limits of what can be learned in the presence of an evasion adversary. In particular, we extend the Probably Approximately Correct (PAC)-learning framework to account for the presence of an adversary. We first define corrupted hypothesis classes which arise from standard binary hypothesis classes in the presence of an evasion adversary and derive the Vapnik-Chervonenkis (VC)-dimension for these, denoted as the adversarial VC-dimension. We then show that sample complexity upper bounds from the Fundamental Theorem of Statistical learning can be extended to the case of evasion adversaries, where the sample complexity is controlled by the adversarial VC-dimension. We then explicitly derive the adversarial VC-dimension for halfspace classifiers in the presence of a sample-wise norm-constrained adversary of the type commonly studied for evasion attacks and show that it is the same as the standard VC-dimension. Finally, we prove that the adversarial VC-dimension can be either larger or smaller than the standard VC-dimension depending on the hypothesis class and adversary, making it an interesting object of study in its own right.

## 1 Introduction

Machine learning (ML) has become ubiquitous due to its impressive performance in domains as varied as image recognition [48, 71], natural language and speech processing [22, 24, 39], game-playing [11, 57, 70] and aircraft collision avoidance [42]. However, its ubiquity provides adversaries with both opportunities and incentives to develop strategic approaches to fool machine learning systems during both training (poisoning attacks) [8, 41, 58, 65] and test (evasion attacks) [7, 15, 34, 55, 56, 61, 75] phases. Our focus in this paper is on evasion attacks targeting the test phase, particularly those based on adversarial examples which add imperceptible perturbations to the input in order to cause misclassification. A large number of adversarial example-based evasion attacks have been proposed against supervised ML algorithms used for image classification [7, 15, 18, 34, 61, 75], object detection [20, 52, 80], image segmentation [3, 31], speech recognition [16, 83] as well as other tasks [21, 36, 43, 82]; generative models for image data [46] and even reinforcement learning algorithms [40, 47]. These attacks have been carried out in black-box [6, 10, 19, 51, 59, 60, 75] as well as in physical settings [27, 49, 69, 73].

To counter these attacks, defenses based on the ideas of adversarial training [34, 53, 76], input denoising through transformations [5, 23, 26, 67, 81], distillation [63], ensembling [1, 4, 74] and feature nullification [77] have been proposed. A number of *detectors* [30, 31, 33, 35, 54] for adversarial examples have also been proposed. However, recent work [12–14] has demonstrated that modifications to existing attacks are sufficient to generate adversarial examples that bypass both defenses

and detectors. In light of this burgeoning arms race, defenses that come equipped with theoretical guarantees on robustness have recently been proposed [45, 64, 72]. These have been demonstrated for neural networks with up to four layers.

In this paper, we take a more fundamental approach to understanding the robustness of supervised classification algorithms by extending well-understood results for supervised batch learning in statistical learning theory. In particular, we seek to understand the sample complexity of Probably Approximately Correct (PAC)-learning in the presence of adversaries. This was raised as an open question for halfspace classifiers by Schmidt et al. [66] in concurrent work which focused on the sample complexity needed to learn *specific distributions*. We close this open question by showing that the sample complexity of PAC-learning when the hypothesis class is the set of halfspace classifers *does not increase* in the presence of adversaries bounded by convex constraint sets. We note that the PAC-learning framework is distribution-agnostic, i.e. it is a statement about learning given independent, identically distributed samples from *any* distribution over the input space. We show this by first introducing the notion of *corrupted hypothesis classes*, which arise from standard hypothesis classes in the binary setting in the presence of an adversary. Now, in the standard PAC learning setting, i.e. adversaries present, the Vapnik-Chervonenkis (VC)-dimension is a way to characterize the 'size' of a hypothesis class which allows for the determination of which hypothesis classes are learnable and with how much data (i.e. sample complexity). In the adversarial setting, we introduce the notion of *adversarial VC-dimension* which is the VC-dimension of the corrupted hypothesis class. With these definitions in place, we can then prove sample complexity upper bounds from the Fundamental Theorem of Statistical Learning in the presence of adversaries that utilize the adversarial VC-dimension.

In this setting, we explicitly compute the adversarial VC-dimension for the hypothesis class comprising all halfspace classifiers, which then directly gives us the sample complexity of PAC-learning in the presence of adversaries. This hypothesis class has a VC-dimension of 1 more than the dimension of the input space when no adversaries are present. We prove that this does not increase in the presence of an adversary, i.e., *the adversarial VC-dimension is equal to the VC-dimension for the hypothesis class comprising all halfspace classifiers*. Our result then raises the question: is the adversarial VC-dimension always equal to the standard VC-dimension? We answer this question in the negative, by showing explicit constructions for hypothesis classes and adversarial constraints for which the adversarial VC-dimension can be arbitrarily larger or smaller than the standard VC-dimension.

**Contributions:** In this paper, we are the first to provide sample complexity bounds for the problem of PAC-learning in the presence of an evasion adversary. We show that an analog of the VC-dimension which we term the adversarial VC-dimension allows us to establish learnability and upper bound sample complexity for the case of binary hypothesis classes with the 0-1 loss in the presence of evasion adversaries. We explicitly compute the adversarial VC-dimension for halfspace classifiers with adversaries with standard $\ell_p$ ($p \geq 1$) distance constraints on adversarial perturbations, and show that it matches the standard VC-dimension. This implies that the sample complexity of PAC-learning does not increase in the presence of this type of adversary. We also show that this is not always the case by constructing hypothesis classes where the adversarial VC-dimension is arbitrarily larger or smaller than the standard one.

## 2 Adversarial agnostic PAC-learning

In this section, we set up the problem of agnostic PAC-learning in the presence of an evasion adversary which presents the learner with adversarial test examples but does not interfere with the training process. We also define the notation for the rest of the paper and briefly explain the connections between our setting and other work on adversarial examples.

We summarize the basic notation in Table 1. We extend the agnostic PAC-learning setting introduced by Haussler [37] to include an evasion adversary. In our extension, the learning problem is as follows. There is an unknown $P \in \mathbb{P}(\mathcal{X} \times \mathcal{C})$.[1] The learner receives labeled training data $(\mathbf{x}, \mathbf{c}) = ((x_0, c_0), \ldots, (x_{n-1}, c_{n-1})) \sim P^n$ and must select $\hat{h} \in \mathcal{H}$. The evasion adversary receives a labeled natural example $(x_{\text{Test}}, c_{\text{Test}}) \sim P$ and selects $y \in N(x_{\text{Test}})$, the set of adversarial examples in the

| Symbol | Usage |
|---|---|
| $\mathcal{X}$ | Space of examples |
| $\mathcal{C} = \{-1, 1\}$ | Set of classes |
| $\mathcal{H} \subseteq (\mathcal{X} \to \mathcal{C})$ | Set of hypotheses (labelings of examples) |
| $\ell(c, \hat{c}) = \mathbb{1}(c \neq \hat{c})$ | 0-1 loss function |
| $R \subseteq \mathcal{X} \times \mathcal{X}$ | Binary nearness relation |
| $N(x) = \{y \in \mathcal{X} : (x, y) \in R\}$ | Neighborhood of nearby adversarial examples |

Table 1: Basic notation used

neighborhood of $x_{\text{Test}}$. The adversary gives $y$ to the learner and the learner must estimate $c_{\text{Test}}$. Their performance is measured by the 0-1 loss, $\ell(c_{\text{Test}}, \hat{h}(y))$.

The neighborhoods $N(x)$ of possible adversarial samples are generated by the binary nearness relation $R$: $N(x) = \{y \in \mathcal{X} : (x, y) \in R\}$. We require $N(x)$ to be nonempty so some choice of $y$ is always available.[2] When $R$ is the identity relation, $I_{\mathcal{X}} = \{(x, x) : x \in \mathcal{X}\}$, the neighborhood is $N(x) = \{x\}$ and $y = x_{\text{Test}}$, giving us the standard problem of learning without an adversary. If $R_1, R_2$ are nearness relations and $R_1 \subseteq R_2$, $R_2$ represents a stronger adversary. One way to produce a relation $R$ is from a distance $d$ on $\mathcal{X}$ and an adversarial budget constraint $\epsilon$: $R = \{(x, y) : d(x, y) \leq \epsilon\}$. This provides an ordered family of adversaries of varying strengths and has been used extensively in previous work [17, 34, 66].

Now, we define the Adversarial Expected and Empirical Risks to measure the learner's performance in the presence of an evasion adversary.

**Definition 1** (Adversarial Expected Risk). *The learner's risk under the true distribution in the presence of an adversary constrained by the relation $R$ is*

$$L_P(h, R) = \mathbb{E}_{(x,c) \sim P}[\max_{y \in N(x)} \ell(h(y), c)].$$

Let $h^* = \operatorname{argmin}_{h \in \mathcal{H}} L_P(h, R)$. Then, learning is possible if there is an algorithm that, with high probability, gives us $\hat{h}_n$ such that $L_P(\hat{h}_n) - L_P(h^*) \to 0$.

Since the learner does not have access to the true distribution $P$, it is approximated with the distribution of the empirical random variable, which is equal to $(x_i, c_i)$ with probability $1/n$ for each $i \in \{0, \ldots, n-1\}$.

**Definition 2** (Adversarial Empirical Risk Minimization (AERM)). *The adversarial empirical risk minimizer* $\text{AERM}_{\mathcal{H}, R} : (\mathcal{X} \times \mathcal{C})^n \to (\mathcal{X} \to \mathcal{C})$ *is defined as*

$$\text{AERM}_{\mathcal{H}, R}(\mathbf{x}, \mathbf{c}) = \operatorname*{argmin}_{h \in \mathcal{H}} L_{(\mathbf{x}, \mathbf{c})}(h, R),$$

*where $L_{(\mathbf{x}, \mathbf{c})}$ is the expected loss under the empirical distribution.*

Clearly, a stronger adversary leads to worse performance for the best possible classifier and in turn, worse performance for the learner.

**Lemma 1.** *Let* $A : (\mathcal{X} \times \mathcal{C})^n \to (\mathcal{X} \to \mathcal{C})$ *be learning algorithm for a hypothesis class $\mathcal{H}$. Suppose $R_1, R_2$ are nearness relations and $R_1 \subseteq R_2$. For all $P$,*

$$\inf_{h \in \mathcal{H}} L_P(h, R_1) \leq \inf_{h \in \mathcal{H}} L_P(h, R_2).$$

*For all $P$ and all $(\mathbf{x}, \mathbf{c})$,*

$$L_P(A(\mathbf{x}, \mathbf{c}), R_1) \leq L_P(A(\mathbf{x}, \mathbf{c}), R_2). \tag{1}$$

*Proof.* For all $h \in \mathcal{H}$,

$$\{(x, c) : \exists y \in N_1(x) \,.\, h(y) \neq c\} \subseteq \{(x, c) : \exists y \in N_2(x) \,.\, h(y) \neq c\}$$

so $L_P(h, R_1) \leq L_P(h, R_2)$. $\qquad\square$

In other words, if we design a learning algorithm for an adversary constrained by $R_2$, its performance against a weaker adversary is better. Crucially, note that the algorithm A on both sides of the inequality in Eq. 1 must be the same for the inequality to hold.

While it is clear that the presence of an adversary leads to a decrease in the optimal performance for the learner, we are now interested in the effect of an adversary on *sample complexity*. If we add an adversary to the learning setting defined in Definition 3, what happens to the gap in performance between the optimal classifier and the learned classifier?

**Definition 3** (Learnability and Sample Complexity). *A hypothesis class $\mathcal{H}$ is learnable by empirical risk minimization in the presence of an evasion adversary constrained by $R$ if there is a function $m_{\mathcal{H},R} : (0,1)^2 \to \mathbb{N}$ (the sample complexity) with the following property. For all $0 < \delta < 1$ and $0 < \epsilon < 1$, all $n \geq m_{\mathcal{H},R}(\delta, \epsilon)$, and all $P \in \mathbb{P}(\mathcal{X} \times \mathcal{C})$,*

$$P^n\big[\big\{(\mathbf{x}, \mathbf{c}) : L_P(\mathrm{AERM}_{\mathcal{H},R}(\mathbf{x}, \mathbf{c}), R) - \inf_{h \in \mathcal{H}} L_P(h, R) \leq \epsilon\big\}\big] \geq 1 - \delta.$$

## 3 Adversarial VC-dimension and sample complexity

In this section, we first describe the notion of corrupted hypotheses, which arise from standard hypothesis classes with the addition of an adversary. We then compute the VC-dimension of these hypotheses, which we term the adversarial VC-dimension and use it to prove the sample complexity upper bounds learning in the presence of an evasion adversary.

### 3.1 Corrupted hypotheses

The presence of an evasion adversary forces us to learn using a *corrupted* set of hypotheses. Unlike ordinary hypotheses that always output some class, these also output the special value $\perp$ that means "always wrong". This corresponds to the adversary being able to select whichever output does not match $c$. This is illustrated in Figure 1.

Let $\widetilde{\mathcal{C}} = \{-1, 1, \perp\}$, where $\perp$ is the special "always wrong" output. We can combine the information in $\mathcal{H}$ and $R$ into a single set $\widetilde{\mathcal{H}} \subseteq (\mathcal{X} \to \widetilde{\mathcal{C}})$ by defining the following mapping where $\kappa_R : (\mathcal{X} \to \mathcal{C}) \to (\mathcal{X} \to \widetilde{\mathcal{C}})$ and $\kappa_R(h) : \mathcal{X} \to \widetilde{\mathcal{C}}$:

$$\kappa_R(h) = x \mapsto \begin{cases} -1 & \forall y \in N(x) : h(y) = -1 \\ 1 & \forall y \in N(x) : h(y) = 1 \\ \perp & \exists y_0, y_1 \in N(x) : h(y_0) = -1, h(y_1) = 1. \end{cases}$$

The corrupted set of hypotheses is then $\widetilde{\mathcal{H}} = \{\kappa_R(h) : h \in \mathcal{H}\}$.

We note that the equivalence between learning an ordinary hypothesis with an adversary and learning a corrupted hypothesis without an adversary allows us to use standard proof techniques to bound the sample complexity.

**Lemma 2.** *For any nearness relation $R$ and distribution $P$,*

$$L_P(h, R) = L_P(\kappa_R(h), I_{\mathcal{X}}).$$

*Proof.* Let $\tilde{h} = \kappa_R(h)$. For all $(x, c)$,

$$\max_{y \in N(x)} \ell(h(y), c) = \mathbb{1}(\exists y \in N(x) . h(y) \neq c) = \mathbb{1}(\tilde{h}(x) \neq c) = \max_{y \in \{x\}} \ell(\tilde{h}(y), c)$$

so $L_P(h, R) = \mathbb{E}[\max_{y \in N(x)} \ell(h(y), c)] = \mathbb{E}[\max_{y \in \{x\}} \ell(\tilde{h}(y), c)] = L_P(\tilde{h}, I_{\mathcal{X}})$. $\qquad\square$

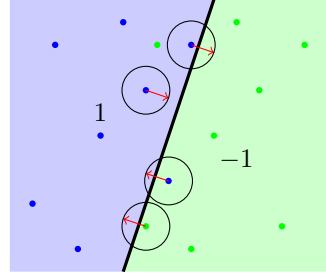

(a) Optimal evasion attacks against halfspace classifiers with *circles* representing the nearness relation $R$

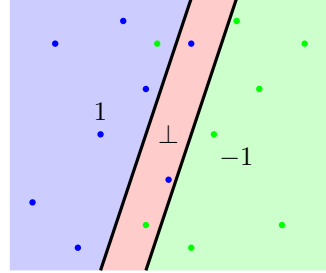

(b) Corrupted halfspace classifier

Figure 1: **Combining the family of hypotheses with the nearness relation $R$.** The top figure depicts some $h \in \mathcal{H}$ and the bottom shows $\kappa_R(h) \in \widetilde{\mathcal{H}}$.

Now, we define the loss class that arises from a hypothesis class. Each element of a loss class is a function produced from the combination of the loss function with a classifier function. The loss function derived from a classifier $h$ is $\lambda(h) : \mathcal{X} \times \mathcal{C} \to \{0, 1\}$, $\lambda(h) = (y, c) \mapsto \ell(c, h(y))$. Thus we have the higher order function $\lambda : (\mathcal{X} \to \widetilde{\mathcal{C}}) \to (\mathcal{X} \times \mathcal{C} \to \{0, 1\})$. Define $\mathcal{F}, \widetilde{\mathcal{F}} \subseteq (\mathcal{X} \times \mathcal{C} \to \{0, 1\})$ to be the loss classes derived from $\mathcal{H}$ and $\widetilde{\mathcal{H}}$ respectively: $\mathcal{F} = \{\lambda(h) : h \in \mathcal{H}\}$ and $\widetilde{\mathcal{F}} = \{\lambda(\tilde{h}) : \tilde{h} \in \widetilde{\mathcal{H}}\}$.

Using this concept and notation, we can restate a standard result from the Rademacher complexity approach to proving sample complexity bounds.

**Lemma 3** ( [68] Theorem 26.5). *Let $\hat{f} = \lambda(\kappa_R(\mathrm{AERM}_{\mathcal{H}, R}(\mathbf{x}, \mathbf{c})))$. With probability $1 - \delta$,*

$$\mathbb{E}_P(\hat{f}(x, c)) - \inf_{f \in \widetilde{\mathcal{F}}} \mathbb{E}_P(f(x, c)) \leq 2R(\widetilde{\mathcal{F}}(\mathbf{x}, \mathbf{c}))) + \sqrt{\frac{32 \log(4/\delta)}{n}}$$

*where*

$$\widetilde{\mathcal{F}}(\mathbf{x}, \mathbf{c}) = \{(\tilde{f}(x_0, c_0), \ldots, \tilde{f}(x_{n-1}, c_{n-1})) : \tilde{f} \in \widetilde{\mathcal{F}}\}$$
$$R(\mathcal{T}) = \frac{1}{n2^n} \sum_{s \in \{-1, 1\}^n} \sup_{t \in \mathcal{T}} s^T t.$$

In the case of non-adversarial learning, i.e. $R = I_{\mathcal{X}}$, this gives a familiar upper bound on $L_P(\hat{h}) - \inf_{h \in \mathcal{H}} L_P(h)$. To get the correct generalization for the adversarial case, we needed to work with the loss class rather than the hypothesis class.

## 3.2 Adversarial VC-dimension: VC-dimension for corrupted hypothesis classes

We begin by providing two equivalent definitions of a shattering coefficient, which we use to determine VC-dimension for standard binary hypothesis classes and adversarial VC-dimension for their corrupted counterparts.

**Definition 4** (Equivalent shattering coefficient definitions). *The $i^{th}$ shattering coefficient of a family of binary classifiers $\mathcal{H} \subseteq (\mathcal{X} \to \mathcal{C})$ is $\sigma(\mathcal{H}, i) = \max_{\mathbf{y} \in \mathcal{X}^i} |\{(h(y_0), \ldots, h(y_{i-1})) : h \in \mathcal{H}\}|$.*

*The alternate definition of shattering in terms of the loss class $\mathcal{F} \subseteq (\mathcal{X} \times \mathcal{C} \to \{0, 1\})$ is*
$$\sigma'(\mathcal{F}, i) = \max_{(\mathbf{y}, \mathbf{c}) \in \mathcal{X}^i \times \mathcal{C}^i} |\{(f(y_0, c_0), \ldots, f(y_{i-1}, c_{i-1})) : f \in \mathcal{F}\}|.$$

Note that these two definitions are indeed equivalent. If $\mathcal{F}$ achieves $k$ error patterns on $(\mathbf{y}, \mathbf{c})$, then $\mathcal{H}$ achieves $k$ classification patterns on $\mathbf{y}$. If $\mathcal{H}$ achieves $k$ classification patterns on $\mathbf{y}$, then $\mathcal{F}$ achieves $k$ error patterns on $(\mathbf{y}, \mathbf{c})$ for any choice of $\mathbf{c}$. Thus, $\sigma(\mathcal{H}, i) = \sigma'(\mathcal{F}, i)$.

The ordinary VC-dimension is then $\mathrm{VC}(\mathcal{H}) = \sup\{n \in \mathbb{N} : \sigma(\mathcal{H}, n) = 2^n\} = \sup\{n \in \mathbb{N} : \sigma'(\lambda(\mathcal{H}), n) = 2^n\}$. The second definition naturally extends to our corrupted classifiers, $\widetilde{\mathcal{H}} \subseteq (\mathcal{X} \to \widetilde{\mathcal{C}})$, because $\lambda(\widetilde{\mathcal{H}}) \subseteq (\mathcal{X} \times \mathcal{C} \to \{0, 1\})$.

**Definition 5** (Adversarial VC-dimension). *The adversarial VC-dimension is*
$$\mathrm{AVC}(\mathcal{H}, R) = \sup\{n \in \mathbb{N} : \sigma'(\lambda(\widetilde{\mathcal{H}}), n) = 2^n\}.$$

These definitions and lemmas can now be combined to obtain a sample complexity upper bound for PAC-learning in the presence of an evasion adversary.

**Theorem 1** (Sample complexity upper bound with an evasion adversary). *For a space $\mathcal{X}$, a classifier family $\mathcal{H}$, and an adversarial constraint $R$, there is a universal constant $C$ such that*

$$m_{\mathcal{H}, R}(\delta, \epsilon) \leq C \frac{d \log(d/\epsilon) + \log(1/\delta)}{\epsilon^2}.$$

*where $d = \mathrm{AVC}(\mathcal{H}, R)$.*

*Proof.* This follows from Lemma 2, Lemma 3, the Massart lemma on the Rademacher complexity of a finite class [68], and the Shelah-Sauer lemma [68]. □

Note that the upper bound on sample complexity can be improved via the chaining technique [25].

# 4 The adversarial VC-dimension of halfspace classifiers

In this section, we consider an evasion adversary with a particular structure, motivated by the practical $\ell_p$ norm-based constraints that are usually imposed on these adversaries in the literature [17, 34]. We then derive the adversarial VC-dimension for halfspace classifiers corrupted by this adversary and show that it remains equal to the standard VC-dimension.

**Definition 6** (Convex constraint on binary adversarial relation). *Let $\mathcal{B}$ be a nonempty, closed, convex, origin-symmetric set.*

*The seminorm derived from $\mathcal{B}$ is $\|x\|_{\mathcal{B}} = \inf\{\epsilon \in \mathbb{R}_{\geq 0} : x \in \epsilon\mathcal{B}\}$ and the associated distance $d_{\mathcal{B}}(x, y) = \|x - y\|_{\mathcal{B}}$.*

*Let $V_{\mathcal{B}}$ be the largest linear subspace contained in $\mathcal{B}$.*

*The adversarial constraint derived from $\mathcal{B}$ is $R = \{(x, y) : y - x \in \mathcal{B}\}$, or equivalently $N(x) = x + \mathcal{B}$.*

Since $\mathcal{B}$ is convex and contains the zero-dimensional subspace $\{\mathbf{0}\}$, $V_{\mathcal{B}}$ is well-defined. Note that this definition of $R$ encompasses all $\ell_p$ bounded adversaries, as long as $p \geq 1$.

**Definition 7.** *Let $\mathcal{H}$ be a family of classfiers on $\mathcal{X} = \mathbb{R}^d$.*

*For an example $x \in \mathcal{X}$ and a classifier $h \in \mathcal{H}$, define the signed distance to the boundary to be*

$$\delta_{\mathcal{B}}(h, x, c) = c \cdot h(x) \cdot \inf_{y \in \mathcal{X}: h(y) \neq h(x)} d_{\mathcal{B}}(x, y)$$

*For a list of examples $\mathbf{x} = (x_0, \ldots, x_{n-1}) \in \mathcal{X}^n$, define the signed distance set to be*

$$D_{\mathcal{B}}(\mathcal{H}, \mathbf{x}, \mathbf{c}) = \{(\delta_{\mathcal{B}}(h, x_0), \ldots, \delta_{\mathcal{B}}(h, x_{n-1})) : h \in \mathcal{H}\}$$

Let $\mathcal{X} = \mathbb{R}^d$ and let $\mathcal{H}$ be the family of halfspace classifiers: $\{(x \mapsto \text{sgn}(a^T x - b)) : a \in \mathbb{R}^d, b \in \mathbb{R}\}$. For simplicity of presentation, we define $\text{sgn}(0) = \perp$, i.e. we consider classifiers that do not give a useful value on the boundary. It is well known that the VC-dimension of this family is $d + 1$ [68]. Our result can be extended to other variants of halfspace classifiers. In Appendix A of our pre-print, we provide an alternative proof that applies to a more general definition.

For halfspace classifiers, the set $D_{\mathcal{B}}(\mathcal{H}, \mathbf{x}, \mathbf{c})$ is easily characterized.

**Theorem 2.** *Let $\mathcal{H}$ be the family of halfspace classfiers of $\mathcal{X} = \mathbb{R}^d$. Let $\mathcal{B}$ be a nonempty, closed, convex, origin-symmetric set. Let $R = \{(x, y) : y - x \in \mathcal{B}\}$. Then $\text{AVC}(\mathcal{H}, R) = d + 1 - \dim(V_{\mathcal{B}})$. In particular, when $\mathcal{B}$ is a bounded $\ell_p$ ball, $\dim(V_{\mathcal{B}}) = 0$, giving $\text{AVC}(\mathcal{H}, R) = d + 1$.*

*Proof.* First, we show $\text{AVC}(\mathcal{H}, R) \leq d + 1 - \dim(V_{\mathcal{B}})$. Define $\|w\|_{\mathcal{B}^*} = \sup_{y \in \mathcal{B}} w^T y$, the dual seminorm associated with $\mathcal{B}$.

For any halfspace classifier $h$, there are $a \in \mathbb{R}^d$ and $b \in \mathbb{R}$ such that $f(x) = \text{sgn}(g(x))$ where $g(x) = a^T x - b$. Suppose that $a^T y = 0$ for all $y \in V_{\mathcal{B}}$. Let $\mathcal{H}'$ be the set of classifiers that are represented by such $a$. For a labeled example $(x, c)$, the signed distance to the boundary is $c(a^T x - b)/\|a\|_{\mathcal{B}^*}$. Any point on the boundary can be written as $x - \epsilon z$ for some $\epsilon \geq 0$ and $z \in \mathcal{B}$. We have $a^T(x - \epsilon z) - b = 0$ so

$$d_{\mathcal{B}}(x, x - \epsilon z) \geq \epsilon = \frac{a^T x - b}{a^T z} \geq \frac{a^T x - b}{\|a\|_{\mathcal{B}^*}}. \tag{2}$$

Because $\mathcal{B}$ is closed, there is some vector $z^* \in \mathcal{B}$ that maximizes $a^T z$. The point $x - \frac{a^T x - b}{\|a\|_{\mathcal{B}^*}} z^*$ is on the boundary, so the inequality 2 is tight.

If we add the restriction $\|a\|_{\mathcal{B}^*} = 1$, each $h \in \mathcal{H}'$ has a unique representation as $\text{sgn} \circ g$. For inputs from the set $\mathcal{G} = \{(a, b) \in \mathbb{R}^{d+1} : \|a\|_{\mathcal{B}^*} = 1, \forall y \in V_{\mathcal{B}} . a^T y = 0\}$, the function $(a, b) \mapsto \delta_{\mathcal{B}}(f, x, c)$ is linear. Thus the function $(a, b) \mapsto (\delta_{\mathcal{B}}(f, x_0, c_0), \ldots, \delta_{\mathcal{B}}(f, x_{n-1}, c_{n-1}))$ is also linear. $\mathcal{G}$ is a subset of a vector space of dimension $d + 1 - \dim V_{\mathcal{B}}$, so

$$\dim(\text{span}(D_{\mathcal{B}}(\mathcal{H}', \mathbf{x}, \mathbf{c}))) \leq d + 1 - \dim V_{\mathcal{B}}$$

for any choices of $\mathbf{x}$ and $\mathbf{c}$.

Now we consider the contribution of the classifiers in $\mathcal{H} \setminus \mathcal{H}'$. These are represented by $a$ such that $a^T y \neq 0$ for some $y \in V_{\mathcal{B}}$, or equivalently $\|a\|_{\mathcal{B}^*} = \infty$. In this case, for all $(x, c) \in \mathbb{R}^d \times \mathcal{C}$, $a^T(x + V_{\mathcal{B}}) = \mathbb{R}$. Thus $x + V_{\mathcal{B}}$ intersects the classifier boundary, $\tilde{h}(x) = \bot$, and the distance from $x$ to the classifier boundary is zero: $\delta_{\mathcal{B}}(h, x, c) = 0$. Thus $D_{\mathcal{B}}(\mathcal{H} \setminus \mathcal{H}', \mathbf{x}, \mathbf{c}) = \{\mathbf{0}\}$, which is already in $\mathrm{span}(D_{\mathcal{B}}(\mathcal{H}', \mathbf{x}, \mathbf{c}))$.

Let $U = \mathrm{span}(D_{\mathcal{B}}(\mathcal{H}, \mathbf{x}, \mathbf{c}))$, let $k = \dim(U)$, and let $n > k$. Suppose that there is a list of examples $\mathbf{x} \in \mathcal{X}^n$ and a corresponding list of labels $\mathbf{c} \in \mathcal{C}^n$ that are shattered by the corrupted classifiers $\widetilde{\mathcal{H}}$. Let $\eta \in \{0, 1\}^n$ be the error pattern achieved by the classifier $h$. Then $\eta_i = \mathbb{1}(\delta_{\mathcal{B}}(h, x_i, c_i) \leq 1)$. In other words, the classification is correct when $c_i$ and $h(x_i)$ have the same sign and the distance from $x_i$ to the classification boundary is greater than 1.

Let $\mathbf{1}$ be the vector of all ones. For each error pattern $\eta$, there is some $h \in \mathcal{H}$ that achieves it if and only if there is some point in $D_{\mathcal{B}}(\mathcal{H}, \mathbf{x}, \mathbf{c}) - \mathbf{1}$ with the corresponding sign pattern.

Since $k < n$, then by the following standard argument, we can find a sign pattern that is not achieved by any point in $U - \mathbf{1}$. Let $w \in \mathbb{R}^n$ satisfy $w^T z = 0$ for all $z \in U$, $w \neq \mathbf{0}$, and $w^T \mathbf{1} \geq 0$. There is a subspace of $\mathbb{R}^n$ of dimension $n - k$ of vectors satisfying the first condition. Since $n > k$, this subspace contains a nonzero vector $u$. At least one of $u$ and $-u$ satisfies the third condition.

If a point $z \in U - \mathbf{1}$ has the same sign pattern as $w$, then $w^T z > 0$. However, we have chosen $w$ such that $w^T z \leq 0$. Thus the classifier family $\mathcal{H}$ does not achieve all $2^n$ error patterns, which contradicts our assumption about $(\mathbf{x}, \mathbf{c})$.

Now we show $\mathrm{AVC}(\mathcal{H}, R) \geq d + 1 - \dim(V_{\mathcal{B}})$ by finding $(\mathbf{x}, \mathbf{c})$ that are shattered by $\widetilde{\mathcal{H}}$.

Let $t = d - \dim(V_{\mathcal{B}})$, $x_0 = \mathbf{0}$, and $(x_1, \ldots, x_t)$ be a basis for the subspace orthogonal to $V_{\mathcal{B}}$. For a subset $S \subseteq \{1, \ldots, t\}$, consider the affine function $g_S(x) = a_S^T x + b_S$ such that $(i \mapsto g(x_i)) = \mathbb{1}(S)$ and $a_S^T x = 0$ for all $x \in V_{\mathcal{B}}$. We would like to use the hyperplane $g_S(x) = \frac{1}{2}$ to achieve the labeling associated with $S$. Let $\delta_S = \max_{x \in \mathcal{B}} a_S^T x$, which is some finite value because $g$ only varies along lines for which $\mathcal{B}$ is bounded. If $\delta_S \geq \frac{1}{2}$ for some $S$, then our current configuration does not work and we must rescale the points to ensure that they can be shattered. Let $\delta = \max(1, 3 \max_{S \subseteq \{1, \ldots, t\}} \delta_S)$. The example list $\delta \cdot (x_0, \ldots, x_t)$ is shattered by $\widetilde{\mathcal{H}}$ for any choice of $\mathbf{c}$. $\square$

## 5 Adversarial VC Dimension can be larger

We have shown in the previous section that the adversarial VC-dimension can be smaller than or equal to the standard VC-dimension. Here, we provide explicit constructions for the counter-intuitive case when adversarial VC-dimension can be arbitrarily larger than the VC-dimension.

**Theorem 3.** *For any $d \in \mathbb{N}$, there is a space $\mathcal{X}$, an adversarial constraint $R \subseteq \mathcal{X} \times \mathcal{X}$, and a hypothesis class $\mathcal{H} : \mathcal{X} \to \mathcal{C}$ such that $\mathrm{VC}(\mathcal{H}) = 1$ and $\mathrm{AVC}(\mathcal{H}, R) \geq d$.*

*Proof.* Let $\mathcal{X} = \mathbb{Z}^d$. Let $\mathcal{H} = \{h_x : x \in \mathcal{X}\}$, where

$$h_x(y) = \begin{cases} 1 & y = x \\ -1 & y \neq x. \end{cases}$$

The VC dimension of this family is 1 because no classifier outputs the labeling $(1, 1)$ for any pair of distinct examples.

Consider the adversary with an $\ell_\infty$ budget of 1. No corrupted classifier will ever output 1, only 0 and $\bot$. Take $\mathbf{x} = (x_0, \ldots, x_{d-1}) \in (\mathbb{Z}^d)^d$:

$$(x_i)_j = \begin{cases} -1 & j = i \\ 1 & j \neq i. \end{cases}$$



Figure 2: The examples $x_0 = (-1, 1)$ and $x_1 = (1, -1)$ are marked with crosses. The function $h_{(0,1)} \in \mathcal{H}$ maps the smaller square to 1 and everything else to $-1$. The corrupted function $\tilde{h}_{(0,1)} \in \widetilde{\mathcal{H}}$ maps the larger square to $\bot$ and everything else to $-1$. Observe that $\tilde{h}_{(0,1)}(x_0) = \bot$ and $\tilde{h}_{(0,1)}(x_1) = -1$.

Now consider the $2^d$ classifiers that are the indicators for $y \in \{0,1\}^{[d]}$: $\tilde{h}_y = \kappa(h_y)$. Observe that

$$\tilde{h}_y(x_i) = \begin{cases} -1 & y_i = 1 \\ \bot & y_i = 0 \end{cases}$$

because if $y_i = 1$ then $d_\infty(x_i, y) = 2$ but if $y_i = 0$ then $d_\infty(x_i, y) = 1$. Thus $(\tilde{h}_y(x_0), \ldots, \tilde{h}_y(x_{d-1}))$ contains $\bot$ at each index that $y$ contains 1. If the examples are all labeled with $-1$, this subset of the corrupted classifier family achieves all $2^d$ possible error patterns. The adversarial VC dimension is at least $d$. $\qquad\square$

Because $\ell(c, -1) \leq \ell(c, \bot)$ for all $c \in \{-1, 1\}$, in the hypothesis class constructed in the proof of Theorem 3, $h_1$ is clearly the best hypothesis. For all $(x, c) \in \mathbb{Z}^d \times \{-1, 1\}$ and all $y \in \{0, 1\}^d$, we have $\ell(c, \tilde{h}_1(x)) \leq \ell(c, \tilde{h}_y(x))$. Thus $h_1$ can be selected without examining any training data and the sample complexity is $m_{\mathcal{H}, R}(\delta, \epsilon) = 0$.

Theorem 3 shows that the addition of an adversary can lead to a significantly weaker upper bound on sample complexity. It does not show that the addition of an adversary can increase the sample complexity and whether this is possible remains an open question.

# 6    Related work and Concluding Remarks

In this paper, we are the first to demonstrate sample complexity bounds on PAC-learning in the presence of an evasion adversary. We now compare with related work and conclude.

## 6.1    Related Work

The body of work on attacks and defenses for machine learning systems is extensive as described in Section 1 and thus we only discuss the closest related work here. We refer interested readers to extensive recent surveys [9, 50, 62] for a broader overview.

**PAC-learning with poisoning adversaries:** Kearns and Li [44] studied learning in the presence of a training time adversary, extending the more benign framework of Angluin and Laird [2] which looked at noisy training data.

**Classifier-specific results:** Wang et al. [78] analyze the robustness of nearest neighbor classifiers while Fawzi et al. [28, 29] analyze the robustness of linear and quadratic classifiers under both adversarial and random noise. Hein and Andriushchenko [38] provide bounds on the robustness of neural networks of up to one layer while Weng et al. [79] use extreme value theory to bound the robustness of neural networks of arbitrary depth. Both these works assume Lipschitz continuous functions. In contrast to our work, all these show how robust a *given classifier* is and do not address the issue of learnability and sample complexity.

**Distribution-specific results:** Schimdt et al. [66] study the sample complexity of learning a mixture of Gaussians as well as Bernoulli distributed data in the presence of $\ell_\infty$-bounded adversaries. For the former, they show that for all classifiers, the sample complexity increases by an order of $\sqrt{d}$, while it only increases for halfspace classifiers for the latter distribution. Gilmer et al. [32] analyze the robustness of classifiers for a distribution consisting of points distributed on two concentric spheres. In contrast to these papers, we prove our results in a *distribution-agnostic* setting.

**Wasserstein distance-based constraint:** Sinha et al. [72] consider a different adversarial constraint, based on the Wasserstein distance between the benign and adversarial distributions. They then study the sample complexity of Stochastic Gradient Descent for minimizing the relaxed Lagrangian formulation of the learning problem with this constraint. Their constraint allows for different samples to be perturbed with different budgets while we study a sample-wise constraint on the adversary.

**Objective functions for robust classifiers:** Raghunathan et al. [64] and Kolter and Wong [45] take similar approaches to setting up a solvable optimization problem that approximates the worst-case adversary in order to carry out adversarial training. Their focus is not on the sample complexity needed for learning, but rather on provable robustness achieved against $\ell_\infty$-bounded adversaries by changing the training objective.

## 6.2 Concluding remarks

While our results provide a useful theoretical understanding of the problem of learning with adversaries, the nature of the 0-1 loss prevents the efficient implementation of Adversarial ERM to obtain robust classifiers. In practice, recent work on adversarial training [34, 53, 76], has sought to improve the robustness of classifiers by directly trying to find a classifier that minimizes the Adversarial Expected Risk, which leads to a saddle point problem [53]. A number of heuristics are used to enable the efficient solution of this problem, such as replacing the 0-1 loss with smooth surrogates like the logistic loss and approximating the inner maximum by a Projected Gradient Descent (PGD)-based adversary [53] or by an upper bound [64]. Our framework now allows for an analysis of the underlying PAC learning problem for these approaches. An interesting direction is thus to find the adversarial VC-dimension for more complex classifier families such as piece-wise linear classifiers and neural networks. Another natural next step is to understand the behavior of convex learning problems in the presence of adversaries, in particular the Regularized Loss Minimization framework.

## Acknowledgments

This work was supported by the National Science Foundation under grants CNS-1553437, CIF-1617286 and CNS-1409415, by Intel through the Intel Faculty Research Award and by the Office of Naval Research through the Young Investigator Program (YIP) Award.

## Footnotes

[1]Formally, we have a sigma algebra $\Sigma \subseteq 2^{\mathcal{X} \times \mathcal{C}}$ of events and $\mathbb{P}(\mathcal{X} \times \mathcal{C})$ is the set of probability measures on $(\mathcal{X} \times \mathcal{C}, \Sigma)$. All hypotheses must be measurable functions relative to $\Sigma$.

[2]Additionally, for all $y \in \mathcal{X}$, $\{x \in \mathcal{X} : (x, y) \in R\}$ should be measurable.

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
