[Reviews · NeurIPS 2018]

Reviewer 1



This paper develops a PAC framework for learning binary functions in the presence of evasion adversaries. Let X be the domain, and suppose we have an unknown function f : X -> {0,1} to be learned, and we also have a hypothesis class H. Moreover, there is a "closeness" relationship defined on pairs in X, so any pair of points in X are either "close" or "far." We also have an unknown distribution P on X. For a hypothesis h, its loss is defined as follows: first we pick a data point x in X according to P. The true label of this point is f(x). If there is a point y "close" to x such that h(y) differs from f(x), then we say x is classified incorrectly under the adversary (the intuition is that, the adversary can slightly perturb the point x into point y to fool the classifier). The loss of a classifier is the probability that a random point x is classified incorrectly under the adversary. Now that we have a loss function, we can ask the standard question in learning theory: given a finite labelled sample from the unknown distribution P, can we pick a hypothesis whose loss is close to being optimal? When there is no adversary, the answer to this question is known, and the sample complexity is characterized by the VC-dimension of the hypothesis class. In this paper, a notion of "adversarial VC-dimension" has been defined, and it is proved that if this dimension is finite, then learning is possible, and a bound is given in terms of this dimension (Theorem 1), which resembles the bound on the sample complexity in terms of VC-dimension in the non-adversarial setting. The proof involves carefully reducing the problem to an appropriate adversary-free learning problem, and using a Rademacher complexity analysis. Then the paper proceeds to understanding the connection between adversarial VC-dimension and standard VC-dimension. An example is given where the former can be arbitrarily larger than the latter. However, it is shown that if the closeness relationship is given by a norm-derived metric, then the VC-dimension of half-space classifiers do not change by adding an adversary. This is rather surprising. I am not familiar with the literature on adversarial learning theory, so I cannot compare the framework and results with the previous ones. Pros: + a clean framework and analysis for binary classification in the presence of adversaries is introduced, which is likely to open doors to new, interesting research. This would be of significance interest to the statistical and learning theory community. + the framework and proofs are novel and mathematically elegant. + the paper is well written. Cons: - the authors have shown that finite adversarial VC-dimension implies learnability. Is the reverse also true? Does learnability under adversary imply finite adversarial VC-dimension? (Note that this is the easy direction in binary classification without adversaries, but it is not addressed in the current paper.) Questions and comments for the authors: * Line 89: add index n for hat(h) (compare with line 104) * 7th line in Table 1: delete "adversarial" * Line 103: "we say" learning is possible if etc. * Line 104: if there is an algorithm that, "upon receiving n training examples", etc. * Line 104: add "as n -> infinity". * Line 105: Rewrite it as: The learner does not have access to the true distribution P, but can approximate it with the empirical distribution. * Line 108: ERM -> AERM * Line 113: be "a" learning algorithm * In Lemma 1, you assume the learning algorithm does not know R. This assumption is without loss of generality, but please explain it in the paper. * Line 131: compute -> define * Display after Line 151: the dot should be replaced with "such that" or ":" * In Definition 4, explain the connection between H and F. * Line 175: C -> tilde(C) * Elaborate on Footnote 3: how can this be improved via chaining? What is the obtained result? * Line 191: "is" is missing * Line 203: what do you mean by "other variants of halfspace classifiers"? * Line 218: inside the formula, replace the dot with "we have" == after reading other reviews and authors' response: please accommodate reviewers' comments and also explain other theoretical work on adversarial learning and how your work differs from them.

Reviewer 2



I have two main concerns with this paper: - the first is on the significance of the result itself. Namely, I agree that provided the definition of what the authors want to capture, then the quantity introduced (adversarial VC dimension) is natural; while the proofs are relatively straightforward, mimicking the non-adversarial VC counterpart, this is not necessarily a bad thing. (Technical or hard is not synonymous of quality.) My issue is rather with the premise itself, as exemplified in Definition 1: after this definition, it is said "Then, learning is possible if [...] L_P(\hat{h})- L_P(h*)-> 0." My feeling is that this sentence is misleading. Learning, in a non-trivial way, is possible when L_P(\hat{h})- L_P(h*)-> 0 AND L_P(h*) is small. Otherwise, without the latter condition (say, if it is 1), then this is essentially saying that there is no hope to learn anything, and characterizes the right number of samples required to learn this nothing. In other terms, I feel the current work seems to address a valid question, but misses a crucial part: that part being "under which assumptions on R, H, and \ell is learning non-trivially in presence of evasion adversaries possible?" - the second issue I have is with the writing. It really feels the authors went out of their way to appear "mathy", sacrificing readability and with no advantage. For instance, the footnotes about measurability: mentioning in passing at the beginning that, as standard,, one assumes measurability of all relevant quantities would be fine; instead, there are two footnotes. to say this, at two occasions. Another one: p.145, writing \kappa_R(h) = x \mapsto [...] instead of \kappa_R(h)(x) = [...] to define the function just makes it harder to parse; the notation is valid, but uncommon enough to keep the reader uneasy. Or, the cllearest example, on p.191: instead of writing "eps >= 0", the authors write "eps \in \mathhbb{R}_{\geq 0}"! More symbols, less clear, no advantage whatsoever. (I could go on: for example, in Theorem 2: "a bounded \ell_p ball" is just "a \ell_p ball" -- why the redundant "bounded"?) Finally, one last comment: in Theorem 1, don't add a footnote saying "This can be improved." Either you prove it, and state it; or you don't, and don't claim it. This is a theorem in your paper, not a remark in passing.

Reviewer 3



This paper considers a new model for PAC learning in the presence of what the paper terms “evasion adversaries”. This setting is one where an attacker changes the examples to generate adversarial examples to make learning difficult. The main contents of the paper include a notion of dimension for this model and learnability results for half-spaces. For half-spaces, the learnability result comes from being able to explicitly analyze the newly-defined “adversarial dimension”, which turns out to be equal to the standard VC-dimension; this type of analysis (as is often the case for other notions of dimension) turns out to be non-trivial and extends to other related classes. Interestingly, at least for me, the new notion of dimension can be larger or smaller than the standard VC dimension. Why this notion could be smaller confused me at first (since VC dimension is used for standard PAC without corruptions), but this notion has an extra parameter, which does away with the contradiction. In the end, I think this is a useful and interesting problem to analyze, but I feel the paper oversells the results. It is not the case, as stated in l.71 “We are the first to provide sample complexity bounds for the problem  of PAC-learning in the presence of an adversary”. Moreover: - For example: given that “adversarial dimension” is also a function of the nearness relation, I think calling it a dimension is a bit too much. - The noise condition is reminiscent of others that should have been better placed in context (eg Tsybakov noise and others). Section 6 does some of job, but is also strangely placed late in the paper. Instead, I think what is interesting is the end-to-end analysis of a particular class in this setting, together with the introduction of techniques that can point to more general progress. But due to some of the criticisms I’ve outlined earlier, my vote is for a “weak accept.” minor: - l.169: shatter->shattering - l.182: Massart lemma -> Massart’s lemma - l.216: “the inequality 1 is tight“ is unclear, as (1) has multiple inequalities. Also the “the” should be removed. - l.229 and later: what are “error patterns”? Do you mean classification pattern? - l.276 “framework of Angluin and Laird which looked at noisy training data” — this is called “classification noise” - VC dimension capitalization is inconsistent, even across section titles (eg section 4 vs 5)